# Deriving the skyrmion Hall angle from skyrmion lattice dynamics

R. Brearton 1,2✉, L. A. Turnbull 3, J. A. T. Verezhak4, G. Balakrishnan 4, P. D. Hatton 3,
G. van der Laan 2✉ & T. Hesjedal 1✉

Magnetic skyrmions are topologically non-trivial, swirling magnetization textures that form lattices in helimagnetic materials. These magnetic nanoparticles show promise as high efficiency next-generation information carriers, with dynamics that are governed by their topology. Among the many unusual properties of skyrmions is the tendency of their direction of motion to deviate from that of a driving force; the angle by which they diverge is a materials constant, known as the skyrmion Hall angle. In magnetic multilayer systems, where skyrmions often appear individually, not arranging themselves in a lattice, this deflection angle can be easily measured by tracing the real space motion of individual skyrmions. Here we describe a reciprocal space technique which can be used to determine the skyrmion Hall angle in the skyrmion lattice state, leveraging the properties of the skyrmion lattice under a shear drive. We demonstrate this procedure to yield a quantitative measurement of the skyrmion Hall angle in the room-temperature skyrmion system FeGe, shearing the skyrmion lattice with the magnetic field gradient generated by a single turn Oersted wire.

[1] Department of Physics, Clarendon Laboratory, University of Oxford, Oxford, UK. [2] Diamond Light Source, Harwell Science and Innovation Campus, Didcot, UK. [3] Department of Physics, Durham University, Durham, UK. [4] Department of Physics, University of Warwick, Coventry, UK. ✉email: Richard.Brearton@physics.ox.ac.uk; Gerrit.vanderLaan@diamond.ac.uk; Thorsten.Hesjedal@physics.ox.ac.uk

**M**agnetic skyrmions (skyrmions hereafter) are topologically robust, localized whirls of magnetization[1–4]—a simulated image of an isolated magnetic skyrmion is shown in Fig. 1a. The topology of a skyrmion gives rise to a Magnus term in its equation of motion[5]. As a result, skyrmions are generally driven at an angle to the direction of applied forces; the relationship between driving force, induced velocity and the skyrmion Hall angle is indicated in Fig. 1b. The stability and mobility of skyrmions have inspired the design and manufacture of a multitude of skyrmion-based devices; despite this, the widespread adoption of magnetic skyrmions in computational schemes is far from reality[6–9]. For the purposes of device manufacturing, the magnitude of the skyrmion Hall angle must be accurately obtained on a material-by-material basis. This is necessary as the skyrmion Hall angle completely dictates the qualitative dynamical properties of skyrmions in a system, and as variations in, e.g., material quality, doping and surface treatment, can alter the many parameters that control the magnitude of the skyrmion Hall angle (such as density of pinning sites, damping and anisotropies) in a manner that is difficult to predict.

For instance, skyrmions for which the Magnus force is large compared to other terms in their equation of motion have been shown to be much less strongly affected by pinning potentials; in this case, the skyrmion Hall angle typically approaches 90°[5]. Furthermore, skyrmions with these properties are expected to enjoy significantly reduced depinning thresholds[10,11], and can be used in device architectures that are incompatible with typical force–velocity relationships[12].

While in many respects a skyrmion Hall angle near to 90° is beneficial, this is not always the case. In device schematics which require skyrmions to be driven along a straight line, it would be desirable for skyrmions to not veer from their path. Non-zero skyrmion Hall angles deflect skyrmions towards the boundary of the structure where they can be trapped or destroyed[13]. This led to a surge of recent interest in skyrmions in antiferromagnetic, synthetic antiferromagnetic and compensated ferrimagnetic materials, which have a skyrmion Hall angle of 0°[14–16].

This difference in qualitative behaviour makes the quantitative determination of the skyrmion Hall angle an important topic; one which has been well tackled for the case of skyrmions in magnetic heterostructures[16–21]. In these systems, skyrmions tend to be sparse, and their controlled nucleation has been well studied[22]. This makes the isolation and manipulation of individual skyrmions possible, allowing the skyrmion Hall angle to be obtained by tracing the real space motion of skyrmions under the influence of a driving force[21]. Such techniques are inapplicable to the study of the skyrmion lattice state, where individual skyrmions cannot readily be labelled or tracked in real space, or where skyrmions are too small to be resolved with the required time resolution.

Here, we report a study of the signatures of lattice dynamics in reciprocal space, showing that the motion of dislocations through a sheared skyrmion lattice forces the lattice to reorient along its direction of motion. The shear applied to the skyrmion lattice also leads to a build-up of strain, which when combined with the lattice reorientation, allows for the unambiguous measurement of the skyrmion Hall angle in reciprocal space.

## Results

**The high-temperature skyrmion system FeGe.** A particularly interesting chiral magnet is the widely studied FeGe, an itinerant helimagnet which hosts sub-100 nm diameter skyrmions near room-temperature[23]. Despite the wealth of research carried out on this material, measurements of the skyrmion Hall angle in FeGe have remained elusive. To experimentally measure the skyrmion Hall angle in FeGe, we set up a shear force with the stray magnetic field from a current-carrying wire, studying the reciprocal space dynamics using resonant elastic X-ray scattering (REXS)[24].

**The effect of shear stress on two-dimensional crystals.** To determine the skyrmion Hall angle from reciprocal space measurements, it is necessary to understand the signature of real space motion in the structure factor. For homogeneous drives, this has been studied in detail, though none of the rich dynamic phases explored in the literature allow for the unambiguous determination of the skyrmion Hall angle from the structure factor alone[10,25]. To move beyond this limitation, it is lucrative to borrow results from the study of crystals under strain. In this case, the properties of a crystal are dictated entirely by its defects and dislocations[26]—an excellent classical example of this is the shear stress required to plastically deform a crystal. In 1926, it was calculated by Frenkel that the shear stress required for plastic deformation of a lattice should be on the order of its shear modulus[27]. In reality, experimental values are 4–8 orders of magnitude smaller[28]. The reason for this stark discrepancy is the existence of dislocations in a real lattice. The shear stress required to plastically deform a real lattice is instead on the order of the

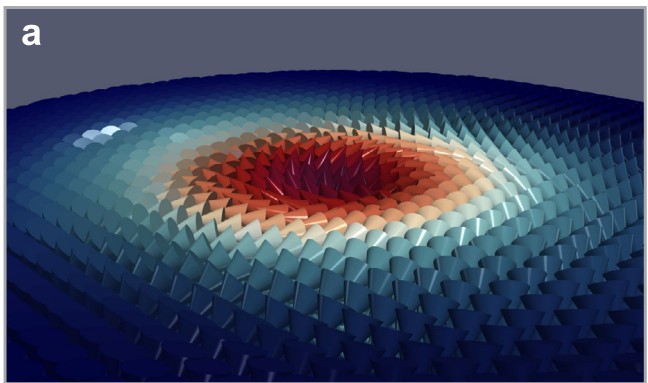

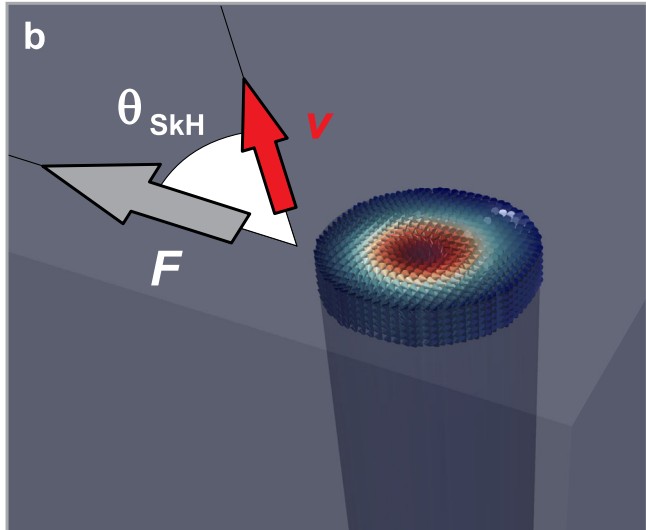

**Fig. 1 Magnetic skyrmions. a** Magnetisation distribution of an isolated magnetic skyrmion. The magnetic moments far from the centre point antiparallel to the central spin, with intermediate moments winding away from the centre in case of a Bloch-type skyrmion (and point radially in case of a Neel-type skyrmion). **b** When a skyrmion is driven by an applied force **F**, it moves with a velocity **v** at an angle $\theta_{SkH}$ to the direction of the applied force. This angle is constrained to lie in the closed interval $[-90°, 90°]$, i.e., skyrmions cannot be driven backwards by a force that acts forwards.

Peierls–Nabarro stress, i.e., the shear stress required to displace a dislocation[29].

As an ideal skyrmion lattice state is two-dimensional (extending as tubes in depth[30,31]), the only form of dislocation present in the lattice are edge dislocations, which typically manifest themselves in hexagonal 2D materials as 5–7 defects[32]. These 5–7 defects can glide in the direction of their Burgers vector for very little energy[28]. The allowed Burgers vectors for 5–7 defects are real primitive lattice vectors, confining defect glides to three special axes. If a shear stress is applied whose direction is not parallel to one of the three axes, motion is possible only by combining glides and climbs. An example visualization of such a defect in a skyrmion lattice, including its Burgers vector, is shown in Fig. 2a.

The climbing motion of 5–7 defects is much more costly than glides, as each climb changes the macroscopic shape of the crystal[28]. If the direction of a shear force is at an angle $\theta$ from one of the glide axes, in order to plastically deform the crystal, there must be $\sin\theta$ defect climbs for every $\cos\theta$ defect glides. As such, the system must spend an energy $E_{\text{Climb}}\sin\theta + E_{\text{Glide}}\cos\theta$ to propagate a defect when relieving strain, where $E_{\text{Climb}} \gg E_{\text{Glide}}$. This creates an effective energy cost $E_{\text{Defect}} = K\sin\theta$ when a 2D crystal is misoriented by an angle $\theta$ to the direction of an applied shear, where $K$ is a constant of proportionality that is proportional to the energy cost of an individual defect climb (which is proportional to the number of particles in the crystal) and the magnitude of the applied shear. To minimize this energy contribution, it is overwhelmingly preferential for a 2D crystal to rotate until it is aligned along the direction of an applied shear. In real space, this would correspond to a shear force acting vertically in Fig. 2a, in which case strain can be easily relieved by propagation of the pictured dislocation along its Burgers vector.

As such, 2D crystals make two key responses to shear forces which are observable in their scattering structure factor. Firstly, the crystal responds by reorienting itself so that the crystal's real lattice vectors are collinear with the shear direction. Secondly, the shear introduces a uniaxial distortion of the structure factor peaks which indicates the direction of motion; this distortion is visible in Fig. 2b. Such distortions are known to occur in sheared colloidal suspensions, and have been recently studied in skyrmion systems[33,34].

**Shearing a skyrmion lattice with a magnetic field gradient.** To apply a shear stress to a skyrmion lattice, the sample was immersed in a non-uniform magnetic field. Assuming that the magnetic field varies slowly over the length scale of a single skyrmion, the Zeeman energy of a skyrmion at position $\vec{R} = (x, y, z)$ can be written as

$$E_{\text{Zeeman}}(\vec{R}) = -\vec{B}(\vec{R}) \cdot I_{\text{Sk}}, \tag{1}$$

where we defined $I_{\text{Sk}} = \int_{A_{\text{Sk}}} \vec{m}_{\text{Sk}}(\vec{r} - \vec{R})dA$. Here, $\vec{m}_{\text{Sk}}$ is the magnetization configuration of a skyrmion and the integral is taken over the area $A_{\text{Sk}}$ bounded by the skyrmion's radius. In the measurements, a constant magnetic field $\vec{B}_0 = B_0\hat{z}$ was applied by permanent magnets, on top of which a perturbative non-uniform magnetic field was generated from a single turn of wire. This gives rise to a field profile which can be written as

$$B_z(y) = B_0 + B_1/y, \tag{2}$$

where the wire lies along the $x$-axis in the Cartesian coordinate system (cf. Oersted wire in Fig. 3). This leads to a force $F_y = I_{\text{Sk}}B_1/y^2$ acting on the skyrmion lattice.

The steady-state motion of a skyrmion under the influence of a driving force is well described by the Thiele equation, given by[35]

$$\vec{F}_M - \alpha\vec{v} = \vec{F}_{\text{ext}} = I_{\text{Sk}}B_1\hat{y}/y^2, \tag{3}$$

where $\alpha$ is the damping coefficient, $\vec{v}$ is the velocity of the skyrmion, and $\vec{F}_{\text{ext}}$ is the external force that acts on the skyrmion. The Magnus force $\vec{F}_M$ can be written as $\vec{F}_M = \vec{G} \times \vec{v}$ and it acts perpendicularly to the direction of the applied velocity, where $\vec{G} = \pm 2\pi N\hat{z}$ is the gyrovector. Here, the topological winding number $N$ is the number of times the skyrmion moments wrap the unit sphere. The angle from the $y$-axis at which skyrmions are driven is given by

$$\theta_{\text{SkH}} = \arctan(G/\alpha). \tag{4}$$

Figure 3 shows velocity fields obtained from solutions of Eq. (3) as a function of $\theta_{\text{SkH}}$. These velocity fields shear the skyrmion lattice along its direction of motion. The direction of the shear can be inferred from the structure factor, and the direction of the force is known; the skyrmion Hall angle is then the angle between the shear and the applied force.

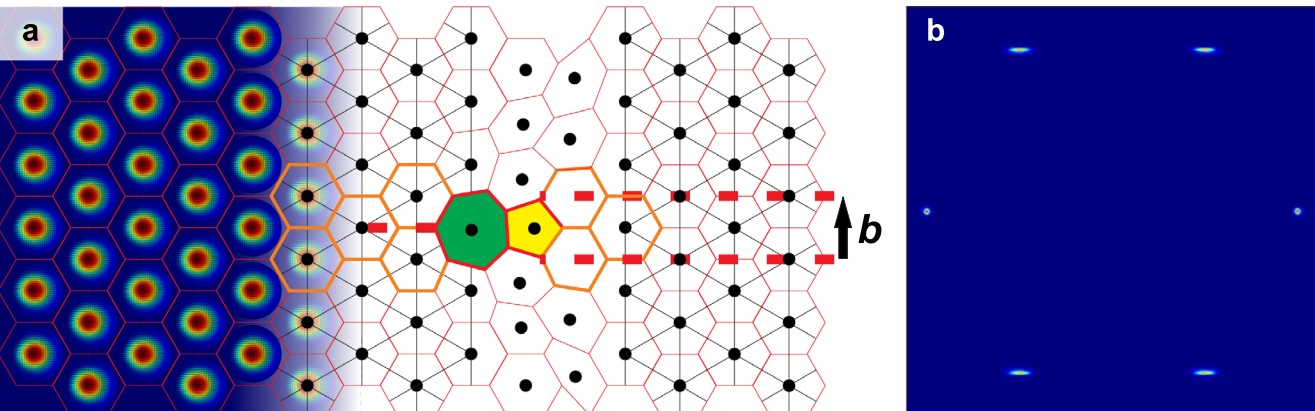

**Fig. 2 Skyrmion lattice defects. a** Visualisation of an edge dislocation in a skyrmion crystal. This dislocation manifests itself as a topological defect, which can be thought of as one skyrmion gaining a nearest neighbour from an adjacent skyrmion; the Voronoi cells surrounding the skyrmions with 5 and 7 nearest neighbours are coloured yellow and green, respectively. This so-called 5–7 defect can glide along its Burgers vector, indicated by **b**, costing very little energy. The Burgers vector of a 5–7 defect must lie along a real lattice basis vector[28]. Were the lattice sheared vertically, strain could be efficiently relieved by propagation of the pictured 5–7 defect along its Burgers vector. **b** Idealized visualization of the diffraction pattern from a strained 2D crystal with a 5–7 defect. Strain in the vicinity of the defect elongates four of the peaks anisotropically: the Burgers vector specifies a unique direction and its corresponding defect breaks the six-fold symmetry of the hexagonal lattice[34].

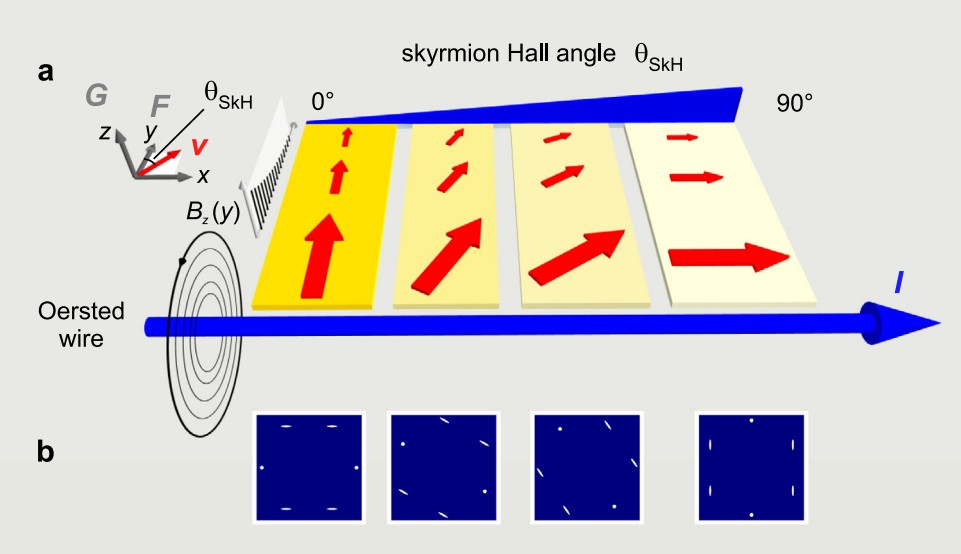

**Fig. 3 Sheared skyrmion lattices in a perturbative linear field gradient for materials with different skyrmion Hall angles. a** The red arrows indicate the magnitude and direction of the induced velocity as a function of distance from the current-carrying Oersted wire and the skyrmion Hall angle $\theta$. **b** For all finite skyrmion Hall angles, the velocity field induced by the field gradient shears the skyrmion lattice along its direction of motion. This shear leads to lattice reorientation and uniaxial peak broadening, indicated in the magnetic structure factors provided beneath their corresponding shear field cartoons. As the direction of the applied force is known, and the direction of the induced motion can be inferred from the structure factor, the skyrmion Hall angle can be acquired by measuring the angle between these two vectors.

**REXS measurements of the skyrmion Hall angle.** To use this approach to measure the skyrmion Hall angle experimentally, REXS was performed in transmission on a thin, $10 \times 10$-$\mu m^2$-sized FeGe lamella with a measured $T_C$ of 273 K[31]. The presence of stray temperature or magnetic field gradients is known to cause the skyrmion lattice to rotate[12,36]. To verify that no such unintended external forces were present in our experimental setup, the motion of the peaks in the REXS diffraction pattern was tracked prior to the application of the shearing magnetic field gradient. This motion was found to be very small, and can be seen by comparing Fig. 4a and b. Subsequently, the response of the skyrmion lattice to the application of a shearing magnetic field gradient was measured; the average of the diffraction patterns obtained after this drive is shown in Fig. 4c. An example video of the response of the skyrmion lattice to the shearing magnetic field is included in the Supplementary Material (Supplementary Video 1), as well as a video of the average response (Supplementary Video 2).

While these data make the reorientation of the skyrmion lattice in response to a shear force strikingly clear, the six-fold symmetry of the acquired diffraction pattern made the unambiguous determination of the skyrmion Hall angle impossible. To break this symmetry, sufficient strain must build up in the skyrmion lattice in response to the shear to broaden the diffraction peaks[34]. Leveraging the reduction in magnetic softness further from $T_C$, the same experiment was performed at 250 K; the results are shown in Fig. 4d. The angle between the labelled direction of motion and the direction of the applied force is the skyrmion Hall angle, and was measured to be $55° \pm 2°$.

## Discussion

This angle is larger than any previous measurement of the skyrmion Hall angle in magnetic multilayer systems. However, this is still far from the theoretical limit of 90°. The primary damping mechanism for skyrmions moving through a metal occurs via interaction with conduction electrons[37]. An insulating skyrmion-hosting material (such as $Cu_2OSeO_3$) may have a skyrmion Hall angle which is closer to the 90° limit, due to greatly reduced damping. The skyrmion Hall angle in such candidate low-dissipation materials can be measured using the presented technique, unlocking the possibility of developing ultra-high mobility devices which take advantage of this unique motion[10–12].

Recent theoretical work suggests that the presence of pinning potentials in real materials should make the skyrmion Hall angle dependent on the magnitude of an applied drive, approaching zero in the low drive limit, while the value given in Eq. (4) is expected to correspond to the high drive limit[10,11]. This hypothesis is supported by experimental evidence, obtained by measuring the skyrmion Hall angle in sputtered magnetic multilayers[17]. The highly energetic nature of the sputtering technique is expected to be the source of the pinning potentials present in these materials systems[17]. As the FeGe lamella used in this experiment was cut from a high-quality single crystal[31], we anticipate that this drive dependence will be suppressed in our sample, and that the angle measured here should be considered to be a lower bound that is close to the theoretical value. A detailed study of the drive dependence of the skyrmion Hall angle in samples cut from a single crystal would make for an interesting topic of future research.

This technique also opens up the prospect of studying the effect of chemical doping on the skyrmion Hall angle in a precise and systematic manner. The ability to fine tune the response of a skyrmion to an applied force represents a significant milestone in the field of skyrmionics, moving the skyrmion Hall angle from an unknown variable to a free parameter. The direction of motion of skyrmions can be further controlled by exploiting the direction in which external forces are applied. While the skyrmion Hall angle is defined as the angle between an applied force and a skyrmion's resultant velocity, the direction in which forces are applied can be complicated by the internal structure of skyrmions. In Eq. (3), the external force $\vec{F}_{ext}$ is set to be equal to the gradient of the Zeeman energy of a skyrmion. When $\vec{F}_{ext}$ is instead equated to the force due to spin-orbit torque from an adjacent heavy metal layer, the Thiele equation becomes

$$\vec{F}_M - \alpha\vec{v} = kR(\phi)\vec{j}_{HM} \qquad (5)$$

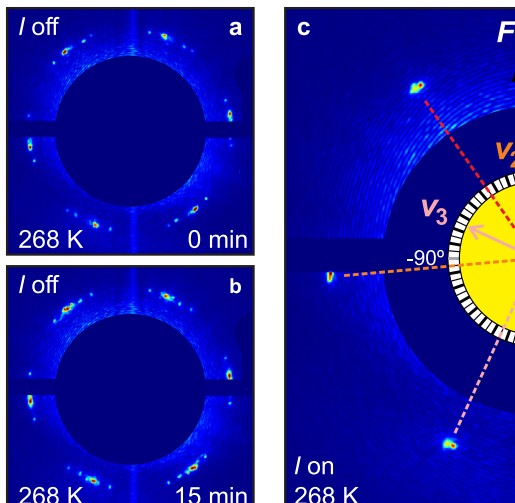

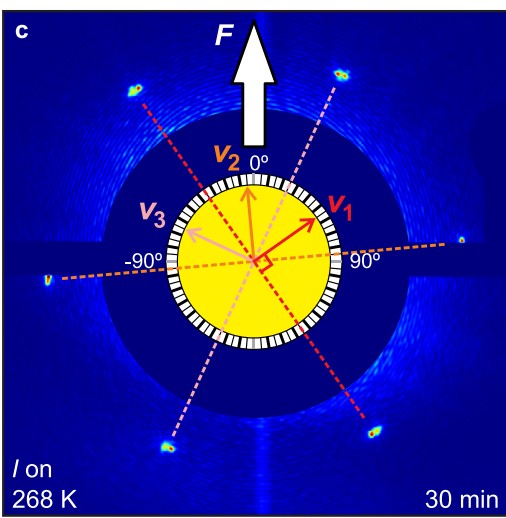

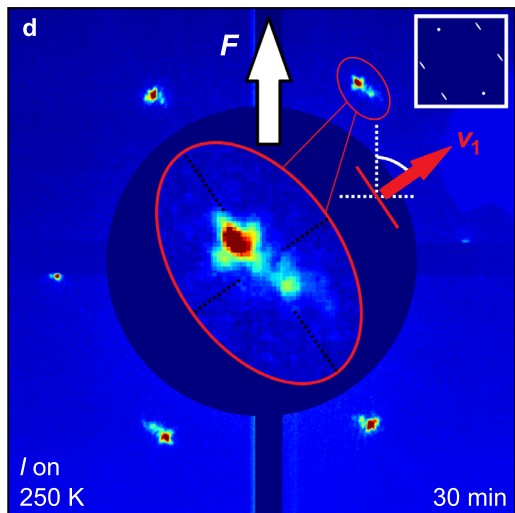

**Fig. 4 Experimental determination of the skyrmion Hall angle of FeGe.** First, the temperature was set to 268 K ($T_C$ =273 K) and a skyrmion-stabilising magnetic field was applied to reach the skyrmion phase pocket. The resonant elastic X-ray scattering (REXS) pattern in **a** shows the typical signature of a skyrmion lattice containing several domains, with multiple sets of six-fold symmetric peaks, see 'Methods' for details. **b** In order to assure stable starting conditions without uncontrolled lattice reorientation, the diffraction experiment was repeated after a sufficient waiting period (15 min), showing a virtually identical pattern. **c** After applying a linear field gradient (I on) for 15 min, the shear force (**F**) introduced by the field gradient drives the skyrmion lattice to completely reorient along its direction of motion, resulting in single, six-fold symmetric spot pattern. A video of the evolution from **a** to **b** to **c** is provided in the Supplementary Materials (Supplementary Video 1). Owing to its six-fold symmetry, there are three possible directions of motion that result in the same diffraction pattern: **v**$_1$ at 55°, **v**$_2$ at −5°, and **v**$_3$ at −65° (note that the skyrmion Hall angle is restricted to [−90°, 90°]). In order to resolve this ambiguity, the stiffness of the skyrmion lattice was tuned from its soft state just below $T_C$ to a harder lattice at a reduced temperature of 250 K, in which strain can build up. **d** The final orientation of the skyrmion lattice is generally the same as for the soft lattice shown in (**c**), however, now the presence of strain leads to a characteristic anisotropic peak broadening (highlighted and magnified, see red ellipse; note that another pair of broadened peaks is masked by the beamstop). The direction of peak broadening is consistent with **v**$_1$ and a Hall angle of 55°, as shown in the simulated REXS pattern in the inset in the top right for which the direction of the shear is 55° away from the direction of the applied force.

where $k$ is a constant of proportionality, $R(\phi)$ is the rotation matrix in the plane spanned by $\vec{F}_M$ and $\vec{v}$, $\phi$ is the helicity angle of a skyrmion and $\vec{j}_{HM}$ is the current density in the heavy metal layer[7]. As Bloch-type skyrmions have a helicity of ±π/2, the force due to spin-orbit torque is normal to $\vec{j}_{HM}$; in this case, the direction of the skyrmion's induced velocity is parallel to $\vec{j}_{HM}$ only when the skyrmion Hall angle is 90°. A detailed discussion of the relationship between current, force and drive direction for Néel- and Bloch-type skyrmions can be found in ref. [7].

In conclusion, we have introduced a novel technique for the straightforward measurement of the skyrmion Hall angle in skyrmion lattice systems. The success of the technique demonstrates that even the properties of a crystal consisting of topologically wound magnetic moments are completely determined by its defects and dislocations, mirroring the discovery of the Peierls–Nabarro stress. We have used this technique to measure the skyrmion Hall angle in a thin lamella of FeGe, finding an angle of 55° ± 2°. This angle far exceeds previous measurements of the skyrmion Hall angle in magnetic multilayer systems, providing verification for previous theoretical work[38]. This giant skyrmion Hall angle opens the door to ultra-low dissipation devices, while the technique allows for the methodical study of the skyrmion Hall angle in the skyrmion lattice state.

## Methods

**Resonant elastic X-ray scattering**. The REXS experiment was carried out in the transmission geometry on a 400-nm-thick, $10 \times 10$ μm$^2$-sized FeGe lamella using a CCD camera[31]. To protect the camera from overexposure in this geometry, a horizontal beamstop was used to block the direct X-ray beam. Nevertheless, the Airy rings which arise from optical diffraction of the beam are extremely bright. These, alongside other contributions from the direct beam, have been masked in all figures to emphasize the magnetic diffraction signal.

**Magnetic gradient field**. To provide the perturbative field, a single turn of 600-μm-diameter Kapton-insulated Cu wire was suspended ~1 mm from the sample, with the core of the wire approximately lying in the plane of the sample surface. An out-of-plane magnetic field gradient of ~1 mT mm$^{-1}$ was mapped out with a Hall probe, obtained by driving 11 A through the wire. The wire was thermally anchored on points, each ~ 10 mm away on both from the sample, and the experiment was performed under ultrahigh vacuum conditions, which minimizes the effects of a perturbative temperature gradient.

Entering the skyrmion phase by field sweeping from the helical state, a roughly randomly oriented skyrmion lattice was obtained. Due to the nature of the small $10 \times 10$ μm$^2$ FeGe samples (prepared by focused-ion-beam milling) there were preferential orientations for the skyrmion lattice in the absence of a driving force, which could be the result of, e.g., a strong shape anisotropy. As a result of this energetic anisotropy, we observed minor natural reorientation of the skyrmion lattice for 1–2 min upon entering the skyrmion pocket in the absence of any external drive. In order to completely rule out any effects due to this reorientation, we found that waiting for 15 min was sufficient. The field gradient was applied after this 15 min pause and the lattice reorientation was measured for a further 15 min. This experiment was performed 15 times at 268 K on a sample of FeGe with a measured $T_C$ of 273 K: four times with an out-of-plane magnetic field of 55 mT, four times at 50 mT, four times at 45 mT, and three times at 60 mT. The reorientation in response to the application of the shearing Oersted field was found to be independent of the out-of-plane magnetic field and was extremely significant in all experiments. The average overall 15 final states is shown in Fig. 4c.

The same experiment was then repeated 24 times at 250 K, eight times each at external magnetic field value of 65, 60, and 55 mT. An average over the final frames obtained after application of the magnetic field gradient for 15 min at 250 K is given in Fig. 4d, with the anisotropic peak broadening indicating the rough direction of motion.

The skyrmion Hall angle was calculated from the data set acquired just below $T_C$ at 268 K, as shown in Fig. 4c. Measuring at this temperature has the advantage that the same information can be obtained in a shorter period of time, as compared to measuring at a lower temperature. As such, the strain induced anisotropy measured in the 250 K experiment was used to infer the approximate direction of motion, while the precise measurement of the skyrmion Hall angle was made using the diffraction spots imaged in Fig. 4c. The quoted uncertainty in the measured value is half of the angular width of the average overall 15 final peak orientations imaged in Fig. 4c.

## Data availability
The authors declare that all other data supporting the findings of this study are available within the paper and its Supplementary information files.

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

## Acknowledgements
The REXS experiments were carried out at beamline I10 at the Diamond Light Source, UK, under proposals MM23784 and MM23451. Financial support by the Engineering and Physical Sciences Research Council (EPSRC) under grant EP/N032128/1 is gratefully acknowledged.

## Author contributions
R.B. and T.H. designed the experiment. R.B., L.T., G.v.d.L. and T.H. carried out the resonant elastic X-ray scattering experiments and analysed the data. J.A.T.V. synthesised the FeGe bulk sample under the supervision of G.B., from which L.T. prepared a thin lamella under the supervision of P.D.H. The manuscript was written by R.B., G.v.d.L. and T.H. with input and comments from all authors.

## Competing interests
The authors declare no competing interests.
