## [Peer Review File · Nature Communications]

Reviewers' Comments:

Reviewer #1:

None

Reviewer #2:

Remarks to the Author:

The manuscript "Deriving the skyrmion Hall angle from skyrmion lattice dynamics" by Brearton et al. reports the determination of the skyrmion Hall angle in reciprocal space by resonant elastic x-ray scattering. The authors are able to extract the angle for bulk FeGe, where skyrmions are stable in a lattice configuration, which does not allow for conventional current-driven measurements of the skyrmion Hall angle. The study is compelling and reports the implementation of an interesting new measurement concept for the skyrmion Hall angle that the authors theorized on last year. That said, there are some issues that I would like to see addressed before I can recommend publication:

1) Line 19: "the angle by which they diverge is a materials constant, known as the skyrmion Hall angle." The equilibrium skyrmion Hall angle is supposedly a material constant, though significant drive dependence has been observed. The authors should clarify this and establish their work in the context of those results.

2) Line 31: "The lack of understanding of the magnitude of the skyrmion Hall angle is a key deterrent for prospective device applications, as this angle completely dictates the qualitative dynamical properties of skyrmions in a system." The magnitude is actually well understood. The drive dependence and its origin are still under debate. If the authors disagree, please clarify.

3) Line 34: "For instance, a skyrmion with a Hall angle approaching 90° would glide around a pinning centre, while a skyrmion with a Hall angle nearer to 0° would be trapped by such a potential well [9]." The cited work is mostly talking about the relative strengths of dissipative and magnus forces, i.e. about the strength of damping in a given system. While this affects the hall angles indeed, the author's conclusion seems generalized and a little misleading (not considering the difference in Hall angles for Bloch and Néel cases, which I will talk more about in point 5). Also, a 90° skyrmion Hall angle is currently undesirable for most applications. The authors may want to give some examples where they envision such a high angle to be advantageous.

4) Line 43: Antiferromagnetic or compensated ferrimagnetic materials, see e.g. Woo 2018.

5) Line 45-47: If the authors state the Hall angle to be well tackled, it would be desirable to see a few more references (e.g. Litzius 2017, Woo 2018, Juge 2019, Litzius 2020, Zeissler 2020). Same holds for the nucleation (Jiang 2015, Finizio 2019). The present manuscript's main advantage over these works is the report of a measurement technique that works in the lattice phase. The authors might want to make this clearer as a selling point. Also, there are several statements of the report of 55° being the largest skyrmion Hall angle observed so far. This is true but out of context. The previous reports were focused on Néel skyrmions, whose Hall angle is 90° rotated with respect to current driven Bloch skyrmions. Looking at the theoretical report the authors published last year they seem not to expect a change in the Hall angle for Néel and Bloch skyrmions in their study. This is a significant difference between their study and current driven experiments and should be pointed out very clearly.

6) Line 77: "As an ideal skyrmion lattice state is two-dimensional (extending as tubes in depth), the only form of dislocation present in the lattice are edge dislocations, which typically manifest themselves in hexagonal 2D materials as 5-7 defects." What would happen if the system contained chiral bobbars as reported by Zheng 2018?

7) Line 138: Would temperature effects limit the study to only certain temperature ranges? The

authors should comment on this in particular.

8) Line 141 and 158: Again, statements of 55° being higher than any previous reported skyrmion Hall angle. This should be put into perspective and the focus of these works should be made clear. The present manuscript does not even mention that Bloch and Néel skyrmions move in different directions when driven by a spin orbit torque (see e.g. Kim 2018) and that this is different for the reported method.

9) Line 142: "However, this is still far from the theoretical limit of 90° ." Several experimental works and the theoretical work by Reichhardt & Reichhardt 2016 showed a significant influence of pinning sites on the skyrmion Hall angle. What is the significance of these effects for the present study? The authors report no change in the skyrmion Hall angle by changing the driving field. Could this be the case because the used drives overcame an additional energy barrier but could not induce free motion of the defects? What tells you that the calculated angle is the limit for your sample? What would be the theoretical skyrmion Hall angle? The authors should clarify this in the manuscript.

Reviewer #3:

Remarks to the Author:

Report on paper: # NCOMMS-20-35818-T by R. Brearton et al.

The remarkable properties of magnetic skyrmions, often associated with their topological character, have aroused enormous interest over the last ten years. Among these, the experimental and theoretical study of a non-Cartesian dynamics leading to a deflection of the skyrmion trajectory perpendicular to the applied force, called the skyrmion Hall effect, is one of the most interesting. Indeed, this phenomenon is a direct signature of the topological nature of skyrmions. Moreover, it is of particular importance in the perspective of using skyrmions in different types of devices. So far, the skyrmion Hall effect has been studied mainly through the measurement of the different velocity components of isolated skyrmions subjected to spin transfer torques as demonstrated among others by W. Jiang et al (Nat. Phys 2017) and also in real time imaging techniques by K. Litzius et al (Nat. Phys. 2017). The study presented in this paper proposes an original and interesting method based on the measurement of the rotation of the peaks in response to field gradient observed in resonant x-ray diffraction to extract the skyrmion Hall angle in the skyrmion lattice phase. In order to demonstrate its efficiency, the authors present a set of experimental data recorder on a skyrmion lattice phase stabilized in FeGe thin film. Before being eventually accepted for publication, I have made here after a list of comments and questions that will have to be addressed.

1) P3, L38: the authors claim that "in many respects a skyrmion Hall angle near to 90° is beneficial". It is not so clear on which aspect a large Hall angle could be an advantage, notably for potential skyrmion based devices. As mentioned by the authors, many attempts have been recently tried to study some systems with cancellation of the skyrmion Hall angle in antiferromagnets but also in ferrimagnetic thin films (not cited in the references) or synthetic antiferromagnets.

2) P4, L77: The authors explain that in ideal skyrmion lattice, edge dislocations are the only types of dislocations and are associated to a specific kind of defects i.e. 5-7 associated to only three special axes for gliding. First for non-specialists, all these statements are only shortly explained and a more detailed introduction, including some schematic diagram (instead of the not so useful fig 1a or in complement to it) would be necessary, in particular to better describe why only these three special axes are expected. Second, as the skyrmion lattices in real systems are not ideal, the authors should comment what would be the consequences and if, for example, other defects could exist with other Burgers vectors and how this could modify the interpretation of the diffraction

patterns?

3) P4, L88: The expression of the energy for the climbs is just introduced and not explained properly. What is exactly the parameter K ? How does it compare to other terms?

4) P5, L112: The Gilbert damping coefficient is a key parameter for the estimation of the skyrmion Hall angle. Do the authors know how large it is for their FeGe film and what is the expected skyrmion Hall angle?

5) P6, L128, Fig4a,b : The authors must explain what are the side peaks present in Fig 4a and 4b. Also, these figures correspond to the diffraction pattern obtained under a perpendicular static field for stabilization of the skyrmion lattice. I guess that other diffraction patterns for lower fields, and hence in the helical phase, have been also recorded. It would be interesting to show them at least as supplementary materials.

6) P6 : L133, Fig. 4c: in Fig 4c, the diffraction diagram recorded in response to a shear force is presented. Even if it shows differences with the ones in Fig 4b, for example much less side peaks, I would not say that there is a striking difference between them as the angles of the six-fold peaks have only slightly changed by a few degrees. It is also not clear why the authors chose to wait for 15 min before recording the diffraction patterns. Is there any physical mechanism related to skyrmion physics associated to such a time scale?

7) P6, L138, Fig 4d : Why the beam stop direction has been changed for the experiments presented in Fig. b,c and the one in Fig 4d? Moreover, the authors use this diffraction pattern to estimate an angle between the direction of the applied force and the supposed direction of motion even though the experimental pattern (fig. 4d) is quite different from the calculated ones (Fig. 3b). In particular, in Fig. 4d, it is not so obvious to differentiate the elongated peaks from the ones that are expected to be punctual. The estimated skyrmion Hall angle is then estimated to be very large + 55° with a relatively small error bar. How this error bar has been determined?

8) P8, L188: The authors must present at least in the supplementary materials the diffraction patterns obtained at different applied perpendicular fields. It would have been also interesting to display the diffraction patterns obtained for different field gradients if the experiments have been performed.

Reviewer reports:

Reviewer #2:

We thank the Reviewer for his/her encouraging words and extremely helpful comments. We appreciate very much his/her effort in straightening out our manuscript.

In the following, we have addressed all the points raised by the Reviewer.

Comment 1: Line 19: *“the angle by which they diverge is a materials constant, known as the skyrmion Hall angle.” The equilibrium skyrmion Hall angle is supposedly a material constant, though significant drive dependence has been observed. The authors should clarify this and establish their work in the context of those results.*

Reply to Comment 1: The Reviewer brings up an important point. Müller & Rosch [PRB 91, 054410 (2015)] and Reichhardt et al. [PRL 114, 217202, (2015)] both independently found that the motion induced by the interaction between skyrmions and pinning potentials should introduce a drive dependence in the skyrmion Hall angle. In the low drive limit the skyrmion Hall angle would approach zero, while in the high drive limit the skyrmion Hall angle would approach its theoretical defect-free value.

The drive dependence of the skyrmion Hall angle was first experimentally observed [Jiang et al., Phys. Rep. 704, 1 (2017)]. Indeed, as drive velocity increased, the skyrmion Hall angle followed suit. Since 2017, many experiments have replicated these results; an elegant demonstration of the importance of this drive dependence can be found in Zeissler et al. [Nat. Commun. 11, 428 (2020)]. In that work, despite the strong dependence of the theoretical skyrmion Hall angle on skyrmion size, no diameter dependence was observable as the skyrmion Hall angle was completely dominated by the presence of pinning potentials.

However, we would like to raise what we believe to be a point of crucial importance: all of the abovementioned studies were carried out on sputtered magnetic multilayers. As sputtering is a relatively violent synthesis technique, in which highly energetic ions are incident on the growth substrate, it is typical for sputtered films to contain a multitude of defects and imperfections that are not present in their single crystalline counterparts. Interestingly, Jiang et al. (2017) associated the drive dependence of their measured skyrmion Hall angle with the very fact that their films were sputtered.

As our samples were cut from a high-quality single crystal of FeGe, we suspect that the drive dependence of the skyrmion Hall angle should be much less pronounced, if at all measurable. In order to verify this hypothesis, a separate, detailed study of the dependence of the skyrmion Hall angle on field gradient magnitude would need to be carried out. This would make for an interesting and topical further study.

We have elaborated on this important point in the discussion, including the following paragraph (complete with references in the updated manuscript):

“Recent theoretical work suggests that the presence of pinning potentials in real materials should make the skyrmion Hall angle dependent on the magnitude of an applied drive, approaching zero in the low-drive limit, while the value given in Eq. (4) is expected to correspond to the high-drive limit [10, 11]. This hypothesis is

supported by experimental evidence, obtained by measuring the skyrmion Hall angle in sputtered magnetic multilayers [17]. The violent nature of the (high energy) sputtering technique is expected to be the source of the pinning potentials present in these materials systems [17]. As the FeGe lamella used in this experiment was cut from a high-quality single crystal [31], we anticipate that this drive dependence will be suppressed in our sample, and that the angle measured here should be considered to be a lower bound that is close to the theoretical value. A detailed study of the drive dependence of the skyrmion Hall angle in samples cut from a single crystal would make for an interesting topic of future research.”

Comment 2: Line 31: *“The lack of understanding of the magnitude of the skyrmion Hall angle is a key deterrent for prospective device applications, as this angle completely dictates the qualitative dynamical properties of skyrmions in a system.” The magnitude is actually well understood. The drive dependence and its origin are still under debate. If the authors disagree, please clarify.*

Reply to Comment 2: We agree that there is a theoretical understanding of the skyrmion Hall angle. However, for the purposes of device manufacturing, the magnitude of the skyrmion Hall angle must be accurately obtained on a material-by-material basis. This is necessary as variations in, e.g., material quality, doping and surface treatment, can alter the many parameters that control the magnitude of the skyrmion Hall angle (such as density of pinning sites, damping and anisotropies) in a manner that is difficult to predict.

Methodical studies of the variation of the skyrmion Hall angle as a function of these materials properties are lacking in the literature. In order to stress the fact that there is a need for specifically a better experimental understanding, we have reworded the sentence:

*“The lack of **experimental data concerning** the skyrmion Hall angle is a key deterrent for prospective device applications, as this angle completely dictates the qualitative dynamical properties of skyrmions in a system.”*

Comment 3: Line 34: *“For instance, a skyrmion with a Hall angle approaching 90° would glide around a pinning centre, while a skyrmion with a Hall angle nearer to 0° would be trapped by such a potential well [9].” The cited work is mostly talking about the relative strengths of dissipative and magnus forces, i.e. about the strength of damping in a given system. While this affects the hall angles indeed, the author’s conclusion seems generalized and a little misleading (not considering the difference in Hall angles for Bloch and Néel cases, which I will talk more about in point 5). Also, a 90° skyrmion Hall angle is currently undesirable for most applications. The authors may want to give some examples where they envision such a high angle to be advantageous.*

Reply to Comment 3: The cited work, “Particle model for skyrmion in metallic chiral magnets: Dynamics, pinning and creep” [Lin et al., PRB 87, 214419 (2013)] contains a fantastic illustration of the qualitative difference between how a skyrmion with a large skyrmion Hall angle and a skyrmion with a small skyrmion Hall angle would interact with a pinning centre or an obstacle. This image (figure 7(a), (b) and (c) in their paper) is pasted below for convenience. Panels (a) and (b) picture the behaviour of a skyrmion with a skyrmion Hall angle approaching 90°, while panel (c) indicates that a skyrmion with a skyrmion Hall angle approaching 0° would get trapped by a pinning centre. The second paragraph in the section entitled

“Comparison between continuum and particle models” contains a detailed comparison between these two limits, with reference to the figure 7(a), (b) and (c).

We agree that the literature is generally pessimistic with regards to the existence of non-zero skyrmion Hall angles, which made our brief statements with regards to the advantages of large skyrmion Hall angles appear rather anaemic. We have modified the main text, including several new references, to reinforce our claim that in certain circumstances a large skyrmion Hall angle could be a strength, not a weakness.

The sentence appended to the end of the first paragraph (excluding references) reads:

Furthermore, skyrmions with large skyrmion Hall angles are expected to enjoy significantly reduced depinning thresholds [10, 11], and can be used in device architectures that are incompatible with typical force-velocity relationships.

Comment 4: Line 43: *Antiferromagnetic or compensated ferrimagnetic materials, see e.g. Woo 2018.*

Reply to Comment 4: Thank you for pointing this out to us. The sentence now reads:

This led to a surge of recent interest in skyrmions in antiferromagnetic, synthetic antiferromagnetic, and compensated ferrimagnetic materials, which have a skyrmion Hall angle of 0° .

Also the reference to Woo et al., Nat. Commun. 10, 1 (2019) has been added.

Comment 5: Line 45-47: *If the authors state the Hall angle to be _well_ tackled, it would be desirable to see a few more references (e.g. Litzius 2017, Woo 2018, Juge 2019, Litzius 2020, Zeissler 2020). Same holds for the nucleation (Jiang 2015, Finizio 2019). The present manuscript’s main advantage over these works is the report of a measurement technique that works in the lattice phase. The authors might want to make this clearer as a selling point. Also, there are several statements of the report of 55° being the largest skyrmion Hall angle observed so far. This is true but out of context. The previous reports were focused on Néel skyrmions, whose Hall angle is 90° rotated with respect to current driven Bloch skyrmions. Looking at the theoretical report the authors published last year they seem not to expect a change in the Hall angle for Néel and Bloch skyrmions in their study. This is a significant difference between their study and current driven experiments and should be pointed out very clearly.*

Reply to Comment 5: We have added the suggested references. In our initial submission, we were limited to only 30 references.

With regards to the skyrmion Hall angle in the special case of spin-transfer torque: spin-transfer torque is a rich effect which applies a force to a skyrmion which is parallel to the current density vector rotated by the helicity angle of the skyrmion. In this experiment, the force applied is due to a magnetic field gradient, which is, to lowest order, given by the spatial derivative of the Zeeman energy of a skyrmion. Skyrmions couple to an external magnetic field via their m_z -component, which contains no helicity information. As a result, the skyrmion Hall angle one would measure from a skyrmion lattice using this technique is independent of the helicity of the constituent skyrmions.

This is a technical detail, but for comparison to the literature we agree that it is important to include this detail in the article. As a result, we have appended the following to the discussion (with references in the main text):

The direction of motion of skyrmions can be further controlled by exploiting the direction in which external forces are applied. The spin-transfer torque effect applies a force to a skyrmion that is rotated by the helicity angle of the skyrmion. As Bloch-type skyrmions have a helicity angle of $\pi/2$, Bloch-type skyrmions with high dissipation (and therefore low skyrmion Hall angles, as dictated by Eq. (4)) will move at roughly 90° to the current direction, emulating the behaviour of a skyrmion with a skyrmion Hall angle of 90° . However, properties such as the depinning threshold of the skyrmion system will be unaffected; the force applied by pinning potentials is central, irrespective of the direction of application of a driving force.

Comment 6: Line 77: “As an ideal skyrmion lattice state is two-dimensional (extending as tubes in depth), the only form of dislocation present in the lattice are edge dislocations, which typically manifest themselves in hexagonal 2D materials as 5-7 defects.” What would happen if the system contained chiral bobbbers as reported by Zheng 2018?

Reply to Comment 6: The Reviewer brings up a very interesting point. The nature of dislocations in three dimensions is of course much richer and complex than in two dimensions. If an abundance of chiral bobbbers was present in our system, it is quite likely that the response of the structure factor to the applied shear would be very different.

However, in this experiment, chiral bobbbers are extremely unlikely to nucleate. In Zheng et al. [Nat. Nanotechnol. 13, 451 (2018)], chiral bobbbers were found only at significantly higher fields and lower temperatures than used in our experiment. To quote the article exactly on the topic of generating chiral bobbbers (ChBs) by field cooling (as in our experiment):

“Although their formation is reproducible, the nucleation of each ChB corresponds to a metastable state in the presence of in-field cooling and is a rare event. In most cases, a change of T_{max} and B_{ext} from their optimal values of 240 K and 200 mT, respectively, results in the formation of a pure multi-skyrmion state.”

Zheng et al. enjoyed success nucleating chiral bobbbers at much lower temperatures (~ 90 K) by cycling a tilted ($\sim 10^\circ$) external magnetic field.

If at all present in our experiment (where $B_{ext} < 100$ mT and $T > 250$ K) their density should be so low that any interference with our experiment would be negligible.

Comment 7: Line 138: *Would temperature effects limit the study to only certain temperature ranges? The authors should comment on this in particular.*

Reply to Comment 7: We found it useful to be able to measure just below T_C , where the response of the system to the drive was most rapid. However, given the significance of the effect at 268 K and at 250 K, we suspect that a wide range of temperatures would show strong effects due to the shearing field gradient.

For our experiment, the $10 \times 10 \mu\text{m}^2$ membrane of FeGe was mounted on the end of a cryostat arm, which expands/contracts for several hours in response to a temperature change. This made changing temperatures extremely time-consuming, so it is impossible for us to make conclusive statements on the effect of temperature changes at this time.

However, we did note that at the lower temperature of 250 K, the structure factor peaks broadened anisotropically, whereas at 268 K this was not resolvable. This signifies that the 250 K state is more smectic than the 268 K state (whose structure factor resembles that of a moving crystal). The implication is that the energy-minimizing 6-fold coordination was more difficult for the system to reach at 250 K than at 268 K, consistent with the fact that magnetic states are significantly more malleable near to T_C , which makes global energetic minima easier to reach. In terms of strain relief, this means that the gliding motion of 5-7 defects was much faster at 268 K than at 250 K.

While this is interesting and merits further study, at this point (with only two data points in temperature) it is impossible for us to make strong comments on the temperature dependence of the effect. However, in the main text we do address the fact that there *is* a non-zero effect due to temperature in the sentence:

Leveraging the reduction in magnetic softness further from T_C , the same experiment was performed at 250 K; the results are shown in Fig. 4d.

Comment 8: Line 141 and 158: *Again, statements of 55° being higher than any previous reported skyrmion Hall angle. This should be put into perspective and the focus of these works should be made clear. The present manuscript does not even mention that Bloch and Néel skyrmions move in different directions when driven by a spin orbit torque (see e.g. Kim 2018) and that this is different for the reported method.*

Reply to Comment 8: As mentioned above, we have now made an explicit comparison between our results and those obtained using spin-transfer torque in the literature. The statement that the measured skyrmion Hall angle exceeds previous measurements of the skyrmion Hall angle is factual, but hopefully brought much more clearly into context by the revised discussion.

Comment 9: Line 142: *“However, this is still far from the theoretical limit of 90° .” Several experimental works and the theoretical work by Reichardt & Reichardt 2016 showed a significant influence of pinning sites on the skyrmion Hall angle. What is the significance of these effects for the present study? The authors report no change in the skyrmion Hall angle by changing the driving field. Could this be the case because the used drives overcame an additional energy barrier but could not induce free motion of the defects? What tells you that the calculated angle is the limit for your sample? What would be the theoretical skyrmion Hall angle? The authors*

should clarify this in the manuscript.

Reply to Comment 9: In response to this comment as well as Comment 1, we have elaborated on the importance of drive dependence in the main text. We reiterate: as this sample is cut from a high quality single crystal of FeGe, we suspect that any drive dependence of the skyrmion Hall angle is greatly reduced when compared to sputtered magnetic multilayers.

→ “*What tells you that the calculated angle is the limit for your sample?*”

This is an important remark – it is of course impossible for us to know that this is the theoretical limit for our sample. It does, however, provide us with a lower bound for the skyrmion Hall angle in our sample. We have revised the main text to incorporate this point by including in the discussion (also see the response to Comment 1):

... we anticipate that this drive dependence will be suppressed in our sample, and that the angle measured here should be considered to be a lower bound that is close to the theoretical value.

→ *What would be the theoretical skyrmion Hall angle?*

The theoretical skyrmion Hall angle is difficult to calculate, as the leading order contribution to the dissipation in metals is due to the non-relativistic coupling between a skyrmion and the electron gas. Previous attempts to quantify this are very approximate (see the section ‘*Estimates*’ in Zang et al. [PRL 107, 136804 (2011)]). In order to quantify this for our sample, we would need to take a series of precise electrical measurements on our 10x10 μm^2 sample, which would pose a significant practical challenge that would only serve to benchmark the theory in Zang (2011). While verification of theoretical understanding is interesting and important, it is not the focus of this work.

Reviewer #3:

We thank the Reviewer for stating that our “study presented in this paper proposes an original and interesting method”, and for suggesting a number of useful changes and additions which we are happy to implement.

Please find below a point-by-point response the comments and questions raised by the Reviewer.

Comment 1: P3, L38: *the authors claim that “in many respects a skyrmion Hall angle near to 90° is beneficial”. It is not so clear on which aspect a large Hall angle could be an advantage, notably for potential skyrmion based devices. As mentioned by the authors, many attempts have been recently tried to study some systems with cancellation of the skyrmion Hall angle in antiferromagnets but also in ferrimagnetic thin films (not cited in the references) or synthetic antiferromagnets.*

Reply to Comment 1: We agree that the literature is generally pessimistic with regards to the existence of non-zero skyrmion Hall angles, which made our brief statements with regards to the advantages of large skyrmion Hall angles appear rather anaemic. We have modified the main text, including several new references, to reinforce our claim that in certain circumstances a large skyrmion Hall angle could be a strength, not a weakness. The newly introduced sentence that ends the first paragraph is listed below:

Furthermore, skyrmions with large skyrmion Hall angles are expected to enjoy significantly reduced depinning thresholds [10, 11], and can be used in device architectures that are incompatible with typical force-velocity relationships [12].

Additionally, we have extended our brief discussion of the literature that focusses on attempts to cancel the skyrmion Hall angle. The updated sentence is pasted below:

This led to a surge of recent interest in skyrmions in antiferromagnetic, synthetic antiferromagnetic and compensated ferrimagnetic materials, which have a skyrmion Hall angle of 0° [14–16].

Comment 2: P4, L77: *The authors explain that in ideal skyrmion lattice, edge dislocations are the only types of dislocations and are associated to a specific kind of defects i.e. 5-7 associated to only three special axes for gliding. First for non-specialists, all these statements are only shortly explained and a more detailed introduction, including some schematic diagram (instead of the not so useful fig 1a or in complement to it) would be necessary, in particular to better describe why only these three special axes are expected. Second, as the skyrmion lattices in real systems are not ideal, the authors should comment what would be the consequences and if, for example, other defects could exist with other Burgers vectors and how this could modify the interpretation of the diffraction patterns?*

Reply to Comment 2: In order to keep the manuscript focused, we took the liberty to only touch upon the topic, providing a reference to an excellent textbook (Hull & Bacon, 2001) that covers the rich subject of crystal defects in detail. However, we hope that we have covered enough details about 5-7 defects to support the claims in our manuscript.

With regards to your second point, it is true that edge dislocations can in general

have Burgers vectors corresponding to any real lattice vector; 5-7 defects represent the special case that the Burgers vectors correspond to a primitive lattice vector. However, the magnitude of a Burgers vector denotes the magnitude of the corresponding lattice distortion. As can be seen by the Voronoi cells surrounding crystal points near the 5-7 defect in figure 2(a), the lattice is already significantly distorted by a Burgers vector of minimal length. Larger Burgers vectors correspond to extreme lattice distortions, which one would not expect to observe in experiment.

This has been confirmed by experiment, in e.g. Pöllath et al. [PRL 118, 207205 (2017)], where great numbers of 5-7 defects were measured using LTEM at domain boundaries. No higher order edge dislocations were reported. In order to strengthen our argument that one need only to consider 5-7 defects we have added a reference to Pöllath et al. (2017) in the main text.

We have also corrected a typing error in this section which could aid with understanding, adding the word primitive: **The allowed Burgers vectors for 5-7 defects are real primitive lattice vectors**

Comment 3: P4, L88: The expression of the energy for the climbs is just introduced and not explained properly. What is exactly the parameter K ? How does it compare to other terms?

Reply to Comment 3: K is a constant of proportionality. As stated in the main text, in order to plastically deform a crystal that is being sheared along a direction that is **not** parallel with a primitive lattice vector, a combination of defect climbs and defect glides is required. Defect glides cost almost no energy at all. Defect climbs involve changing the macroscopic crystal structure, which costs a significant amount of energy.

From trigonometry, as stated in the main text, if the direction of a shear force is at an angle θ from one of the glide axes, in order to plastically deform the crystal, there must be $\sin\theta$ defect climbs for every $\cos\theta$ defect glides. The average energy the system must spend on propagating a defect then looks like:

$$E_{\text{DefectPropagation}} = E_{\text{Glide}} \cos\theta + E_{\text{Climb}} \sin\theta.$$

However, as the energy cost of a glide is negligible, one can just as well write:

$$E_{\text{DefectPropagation}} = E_{\text{Climb}} \sin\theta .$$

Then, the energy the system must spend on relieving strain is proportional to $E_{\text{DefectPropagation}}$, which itself scales with the number of particles in the system, and is also proportional to the magnitude of the strain applied to the lattice (rate of defect propagation is proportional to the applied strain).

We have expanded our discussion of the equation, including the full physical meaning of the constant of proportionality K , following the hopefully clearer argument stated above. The sentences now read:

The climbing motion of 5-7 defects is much more costly than glides, as each climb changes the macroscopic shape of the crystal [28]. If the direction of a shear force is at an angle θ from one of the glide axes, in order to plastically deform the crystal, there must be $\sin\theta$ defect climbs for every $\cos\theta$ defect glides. As such, the system must spend an energy $E_{\text{Climb}} \sin\theta + E_{\text{Glide}} \cos\theta$ to propagate a defect when relieving

strain, where $E_{\text{Climb}} \gg E_{\text{Glide}}$. This creates an effective energy cost $E_{\text{Defect}} = K \sin \theta$ when a 2D crystal is misoriented by an angle θ to the direction of an applied shear, where K is a constant of proportionality related to the energy cost of an individual defect climb (which, in turn, is proportional to the number of particles in the crystal) and the magnitude of the applied shear. To minimise this energy contribution, it is overwhelmingly preferential for a 2D crystal to rotate until it is aligned along the direction of an applied shear. In real space, this would correspond to a shear force acting vertically in Fig. 2a, in which case strain can be easily relieved by propagation of the pictured dislocation along its Burgers vector.

Comment 4: P5, L112: *The Gilbert damping coefficient is a key parameter for the estimation of the skyrmion Hall angle. Do the authors know how large it is for their FeGe film and what is the expected skyrmion Hall angle?*

Reply to Comment 4: The Gilbert damping coefficient, which is relativistic in nature, is indeed a key parameter for determining the skyrmion Hall angle in insulating materials. However, as FeGe is metallic, the leading order contribution to damping for a *moving* skyrmion comes from the non-relativistic coupling between a skyrmion with non-zero velocity and an electron gas [see Zang et al., PRL 107, 136804 (2011)]. Instead of measurements of the Gilbert damping, in order to estimate the skyrmion Hall angle we would need to take precise electrical measurements on our lamella, which given its size would be extremely difficult.

Comment 5: P6, L128, Fig4a,b: *The authors must explain what are the side peaks present in Fig 4a and 4b. Also, these figures correspond to the diffraction pattern obtained under a perpendicular static field for stabilization of the skyrmion lattice. I guess that other diffraction patterns for lower fields, and hence in the helical phase, have been also recorded. It would be interesting to show them at least as supplementary materials.*

Reply to Comment 5: As explained in the methods, figure 4(a) and (b) are averages over many experiments. There are many peaks because, over these experiments, the skyrmion lattice froze in a variety of different orientations.

In the figure caption itself, we previously stated that this is a typical scattering pattern from a “disordered” skyrmion lattice, which we appreciate is not quite technically correct. This has been reworded to:

“The REXS pattern in a shows the typical signature of a skyrmion lattice containing several domains, with multiple sets of six-fold symmetric peaks, see Methods for details.”

This is more technically correct, as an image of the average of scattered intensity from domains separated in time is the same as an image of the scattered intensity from domains separated in space.

With regards to your second comment, a lamella cut from the same crystal has been used for a detailed study of the phase diagram and the magnetic phases. We are referring to this study in the revised version and cite the reference by Birch et al. [Nat. Commun. 11, 1726 (2020)].

For the magnetic phase diagram and real-space images of the magnetic phases, we refer to Ref. [31] for which lamella cut from the same high-quality FeGe single crystal

have been used.

Comment 6: P6: L133, Fig. 4c: in Fig 4c, the diffraction diagram recorded in response to a shear force is presented. Even if it shows differences with the ones in Fig 4b, for example much less side peaks, I would not say that there is a striking difference between them as the angles of the six-fold peaks have only slightly changed by a few degrees. It is also not clear why the authors chose to wait for 15 min before recording the diffraction patterns. Is there any physical mechanism related to skyrmion physics associated to such a time scale?

Reply to Comment 6: First answering the latter part of the question:

“Is there any physical mechanism related to skyrmion physics associated to such a time scale?”

The answer is yes, and there are two physical mechanisms to consider. Firstly, the initial orientation that the skyrmion lattice freezes into is random (biased by details such as sample geometry, field misorientation, anisotropy from the underlying crystal lattice etc.). It is possible that the initial orientation of the skyrmion lattice, including any domains, is not the most energetically preferential configuration of skyrmions. We measured that, over the timescale of ~minutes, minor lattice reorientation takes place as the skyrmion lattice attempts to find a configuration with a lower global energy.

The second effect leading to a timescale on the order of minutes is field gradient driven motion. In a previous proof of concept experiment, a skyrmion lattice was subject to a radial field gradient, which (for all nonzero skyrmion Hall angles) causes the skyrmion lattice to rotate. This rotation took place over the course of minutes – videos of the rotation can be found in the supplementary materials of S. L. Zhang et al., Nat. Commun. 9, 2115 (2018).

In order to control for the first physical mechanism (latent dynamics, minor reorientation), after entering the skyrmion pocket we waited for 15 minutes before driving the skyrmion lattice. We found that the dynamics of the skyrmion lattice in the absence of our drive were negligible. Once we started to drive the skyrmion lattice with a magnetic field gradient we applied the magnetic field gradient for 15 minutes, because we know from Zhang [Nat. Commun. 9, 2115 (2018)] that this is a safe upper bound for the timescale of lattice dynamics induced by a magnetic field gradient.

“I would not say that there is a striking difference between them as the angles of the six-fold peaks have only slightly changed by a few degrees.”

We would have to agree: still figures simply don't do this effect justice – this is why we have attached supplementary videos of the transition from figure 4(b) to 4(c). The first supplementary video shows the result of one experiment, sped up by a factor of roughly 200. In that video, the effect of lattice reorientation in the presence of a field gradient is striking, even if the lattice only reorients by $\sim 10^\circ$.

The experiment shown in the first supplementary video was carried out 14 more times. Each time, the skyrmion lattice started at a different random initial orientation. Each time, the skyrmion lattice rotated to arrive at the same final orientation. The second supplementary video is an average over all these experiments. Of course, in this second video, there is no observable net rotation – the lattice rotates clockwise

about as many times as it rotates counter-clockwise to reach its final orientation.

Finally, we note that the second supplementary video provides justification for our choice of timescale. A small amount of jiggling is noticeable throughout the “current off” phase, corresponding to the latent dynamics of the skyrmion crystal as it tries to find its optimal configuration. Also, once the field gradient is applied (corresponding to “current on” in the videos), it takes several seconds for all of the peaks from each of the experiments to become mutually aligned, meaning that it took a few minutes to guarantee lattice reorientation in the experiment.

In response to this point, we have explicitly referred to the supplementary videos in the caption for figure 4, to encourage the reader to watch for themselves the transition from figure 4(a) to (b) to (c).

Comment 7: P6, L138, Fig 4d : Why the beam stop direction has been changed for the experiments presented in Fig. b,c and the one in Fig 4d? Moreover, the authors use this diffraction pattern to estimate an angle between the direction of the applied force and the supposed direction of motion even though the experimental pattern (fig. 4d) is quite different from the calculated ones (Fig. 3b). In particular, in Fig. 4d, it is not so obvious to differentiate the elongated peaks from the ones that are expected to be punctual. The estimated skyrmion Hall angle is then estimated to be very large $+ 55^\circ$ with a relatively small error bar. How this error bar has been determined?

Reply to Comment 7: We thank the Reviewer for pointing this out, the beam stop direction had not changed. However, during the experiments at lower temperature, an alignment setting had been changed further down the beamline which lowered the intensity of the incident beam. This did not cause a practical problem as magnetic scattering was still very easy to resolve, however, it meant that a streak caused by previous damage to this particular CCD camera was on the same intensity order as magnetic scattering (and was therefore very bright). We have attached an un-edited, slightly cropped, version of figure 4(d) below:

The squares near the centre are caused by the direct beam passing through empty membranes in our sample holder – we had four membranes mounted at the time. The ugly mark on the right edge is the straight beam passing over the permanent magnets. In this larger image, it is hopefully easy to see that the beam stop was actually still horizontal, and that we added a second vertical one to mask the streak coming from detector damage that runs vertically through the screen.

If looking carefully at the image, one can see the remnants of this detector damage in the higher temperature experiments too, but as magnetic intensity was higher the streak shows up as a dark blue.

We decided that it is most reasonable to simply quote the angular width of the largest broadened diffraction spots in figure 4(d) as our greatest measurement error, as mentioned in the main text.

Comment 8: P8, L188: *The authors must present at least in the supplementary materials the diffraction patterns obtained at different applied perpendicular fields. It would have been also interesting to display the diffraction patterns obtained for different field gradients if the experiments have been performed.*

Reply to Comment 8: We thank the Reviewer for pointing this deficiency out to us.

As stated in our response to Comment 5, a repeat of the measurements of the other magnetic phases was not carried out given how precious (and expensive) synchrotron beamtime is. Instead, we refer to the work done by Birch et al. [Nat. Commun. 11, 1726 (2020)] on a very similar lamella cut from the very same crystal by the same person.

It was our intention for this first experiment to extract the skyrmion Hall angle that is as close to the maximum for our sample as possible. We do have data at lower drives, but it is limited and nowhere near as methodically obtained. It was captured while testing the maximum field gradient that we could safely apply before needing to consider effects due to heating. A study of the field-gradient dependence of these data would make for a very interesting follow-up experiment.

In the revised version of our manuscript, we explicitly address the drive dependence of the skyrmion Hall angle in the present literature in the following paragraph (including references in the main document):

“Recent theoretical work suggests that the presence of pinning potentials in real materials should make the skyrmion Hall angle dependent on the magnitude of an applied drive, approaching zero in the low-drive limit, while the value given in Eq. (4) is expected to correspond to the high-drive limit [10, 11]. This hypothesis is supported by experimental evidence, obtained by measuring the skyrmion Hall angle in sputtered magnetic multilayers [17]. The violent nature of the (high energy) sputtering technique is expected to be the source of the pinning potentials present in these materials systems [17]. As the FeGe lamella used in this experiment was cut from a high-quality single crystal [31], we anticipate that this drive dependence will be suppressed in our sample, and that the angle measured here should be considered to be a lower bound that is close to the theoretical value. A detailed study of the drive dependence of the skyrmion Hall angle in samples cut from a single crystal would make for an interesting topic of future research.”

Reviewers' Comments:

Reviewer #2:

Remarks to the Author:

The authors did a thorough job addressing most of my comments. However, there are still a few aspects I would like to be made a little clearer:

on comment 2:

"We agree that there is a theoretical understanding of the skyrmion Hall angle. However, for the purposes of device manufacturing, the magnitude of the skyrmion Hall angle must be accurately obtained on a material-by material basis. This is necessary as variations in, e.g., material quality, doping and surface treatment, can alter the many parameters that control the magnitude of the skyrmion Hall angle (such as density of pinning sites, damping and anisotropies) in a manner that is difficult to predict."

I find this explanation of the key aspects where this technique could help much better than the wording that made it into the manuscript ("The lack of experimental data concerning the skyrmion Hall angle is a key deterrent for prospective device applications...") and advise to rewrite this part with the content of the quote above.

on comment 3:

"Furthermore, skyrmions with large skyrmion Hall angles are expected to enjoy significantly reduced depinning thresholds [10, 11], and can be used in device architectures that are incompatible with typical force-velocity relationships."

And the rebuttal text starting with

"The cited work, 'Particle model for skyrmion in metallic chiral magnets: Dynamics, pinning and creep' [Lin et al., PRB 87, 214419 (2013)] contains a fantastic illustration of the qualitative difference between how a skyrmion with a large skyrmion Hall angle and a skyrmion with a small skyrmion Hall angle..."

The direct relation of skyrmion Hall angle and depinning threshold should be put into a little more perspective. The usual explanation for reduced depinning thresholds is, as Lin et al. state, the interplay (i.e. the ratio) of magnus force and dissipative force, here only measured as the strength of the Gilbert damping. In the clean sample limit this is indeed the skyrmion Hall angle and the statement should be valid in the context of this manuscript. However, it is not necessarily the case for disordered films, where the Hall angle gains additional contributions. Additionally, with respect to the helicity dependence of the skyrmion Hall angle for current induced experiments, the statement should be clarified as low α /high magnus force would mean a lower angle for Bloch skyrmions but still increase the mobility. The authors mention this effect in their discussion later (see also comment below), though I am not convinced that it becomes sufficiently clear to a reader with a different background.

on comment 5:

"The direction of motion of skyrmions can be further controlled by exploiting the direction in which external forces are applied. The spin-transfer torque effect applies a force to a skyrmion that is rotated by the helicity angle of the skyrmion. ..."

I was more interested in the direct comparison of eq. 3+4 to the SOT/STT cases to make the reader aware of the different drive symmetries and thus angles they may observe in current driven experiments for Neel and Bloch cases under STT/SOT. As stated in the paragraph above, I think this aspect can be clarified a little more to avoid confusion.

Reviewer #3:

Remarks to the Author:

In this revised version of the manuscript as well as in their responses, the authors have properly answered to all referee's comments including mine. Thanks to their precisions, I believe that the

manuscript improves and becomes much more clear for the readers. I still have a few comments. First, even if I agree that sputtering deposition is not the smoothest deposition technique but I would not have labelled it as "violent". The impact of this growth technique indeed depends on the materials that is deposited, as for example sputtered CoFeB films show extremely pinning site densities compared to Co ones. Second, because the authors have now introduced a specific argumentation about the drive dependence of the skyrmion Hall effect as well as a citation to several theoretical and experimental references, in order to be more exhaustive, they could probably also add the one published by W. Legrand et al, Nano Lett, 17, 2703 (2017). After these very minor revisions, I believe that this version will fulfill the criteria for publication in Nature Communications.

Reviewer reports:

Reviewer #2:

We thank the Reviewer for acknowledging that we “did a thorough job addressing most of my comments.” We appreciate his/her attention to detail and have made the sections of our manuscript clearer where the Reviewer commented on below.

On comment 2: *“We agree that there is a theoretical understanding of the skyrmion Hall angle. However, for the purposes of device manufacturing, the magnitude of the skyrmion Hall angle must be accurately obtained on a material-by material basis. This is necessary as variations in, e.g., material quality, doping and surface treatment, can alter the many parameters that control the magnitude of the skyrmion Hall angle (such as density of pinning sites, damping and anisotropies) in a manner that is difficult to predict.”*

I find this explanation of the key aspects where this technique could help much better than the wording that made it into the manuscript (“The lack of experimental data concerning the skyrmion Hall angle is a key deterrent for prospective device applications...”) and advise to rewrite this part with the content of the quote above.

Reply to ‘On comment 2’: We have rewritten the mentioned section in the paper to contain the arguments given in our first response to the Reviewer. It now reads as follows:

“For the purposes of device manufacturing, the magnitude of the skyrmion Hall angle must be accurately obtained on a material-by-material basis. This is necessary as the skyrmion Hall angle completely dictates the qualitative dynamical properties of skyrmions in a system, and as variations in, e.g., material quality, doping and surface treatment, can alter the many parameters that control the magnitude of the skyrmion Hall angle (such as density of pinning sites, damping and anisotropies) in a manner that is difficult to predict.”

On comment 3: *“Furthermore, skyrmions with large skyrmion Hall angles are expected to enjoy significantly reduced depinning thresholds [10, 11], and can be used in device architectures that are incompatible with typical force-velocity relationships.” And the rebuttal text starting with “The cited work, ‘Particle model for skyrmion in metallic chiral magnets: Dynamics, pinning and creep’ [Lin et al., PRB 87, 214419 (2013)] contains a fantastic illustration of the qualitative difference between how a skyrmion with a large skyrmion Hall angle and a skyrmion with a small skyrmion Hall angle...”*

The direct relation of skyrmion Hall angle and depinning threshold should be put into a little more perspective. The usual explanation for reduced depinning thresholds is, as Lin et al. state, the interplay (i.e. the ratio) of magnus force and dissipative force, here only measured as the strength of the Gilbert damping. In the clean sample limit this is indeed the skyrmion Hall angle and the statement should be valid in the context of this manuscript. However, it is not necessarily the case for disordered films, where the Hall angle gains additional contributions. Additionally, with respect to the helicity dependence of the skyrmion Hall angle for current induced experiments, the statement should be clarified as low alpha/high magnus force would mean a lower angle for Bloch skyrmions but still increase the mobility. The authors mention this effect in their discussion later (see also comment below), though I am not convinced that it becomes sufficiently clear to a reader with a different background.

Reply to 'On comment 3': In order to keep technical terms to a minimum, we had hoped to avoid branding the $\vec{G} \times \vec{v}$ term in Thiele's equation as the Magnus force. However, as the Reviewer points out, descriptions of more complex force-velocity relationships are more easily made with reference to the Magnus force.

We have therefore modified our introduction to the skyrmion Hall angle in order to accommodate this linguistic change, now making explicit reference to the Magnus force from the beginning of the manuscript. The affected sentence reads:

“The topology of a skyrmion gives rise to a Magnus term in its equation of motion [5]. As a result, skyrmions are generally driven at an angle to the direction of applied forces; the relationship between driving force, induced velocity and the skyrmion Hall angle is indicated in Fig. 1b.”

Note that reference [5] is Lin et al (2013).

Then, the Referee's quoted section of the introduction now reads:

“For instance, skyrmions for which the Magnus force is large compared to other terms in their equation of motion have been shown to be much less strongly affected by pinning potentials; in this case, the skyrmion Hall angle typically approaches 90° [5]. Furthermore, skyrmions with these properties are expected to enjoy significantly reduced depinning thresholds [10, 11], and can be used in device architectures that are incompatible with typical force-velocity relationships [12].”

We now emphasize that significantly reduced depinning thresholds should be expected if the Magnus force is large compared to other terms in the equation of motion of a skyrmion, and that this typically leads to large skyrmion Hall angles. Additionally, the second sentence now uses the phrase “these properties”, as opposed to directly mentioning skyrmion Hall angles that approach 90° , as the statement regarding reduced depinning thresholds relies on the relatively large Magnus force, and the statement regarding device architectures relies on the skyrmion Hall angle. While these changes in nomenclature are subtle, they lead to a more precise manuscript.

We did not change the wording in the next paragraph, beginning “While in many respects a skyrmion Hall angle approaching 90° is beneficial...”, for two reasons:

- Firstly, the statement that a skyrmion Hall angle approaching 90° is beneficial is consistent with the previous paragraph. If the skyrmion Hall angle approaches 90° , certainly the Magnus force is large relative to damping and the depinning threshold will be reduced. (Note, we are not discussing the geometric details of spin-transfer torque applied to a Bloch-type skyrmion with a skyrmion Hall angle of 90° here. Such a specific topic is better placed in, and can in fact be found in, the discussion.)
- Secondly, we did not want to replace every reference to skyrmion Hall angles with discussions of relative magnitudes of forces (namely the Magnus and dissipative forces). This would make the introduction overly technical and difficult to read.

Additionally, our formulation of Thiele's equation has been modified. It is now presented as:

$$\vec{F}_M - \alpha \vec{v} = \vec{F}_{ext} = I_{sk} B_1 \hat{y} / y^2$$

Where the presence of $I_{sk} B_1 \hat{y} / y^2$ in the equality is contextually very clear from the content of the preceding paragraphs, where that term is derived. The paragraph that

follows Thiele's equation now reads:

“... where α is the damping coefficient, \vec{v} is the velocity of the skyrmion, and \vec{F}_{ext} is the external force that acts on the skyrmion. The Magnus force \vec{F}_M can be written as $\vec{F} = \vec{G} \times \vec{v}$ and acts perpendicularly to the direction of the applied velocity, where $\vec{G} = \pm 2\pi N \hat{z}$ is the gyrovector. Here, the topological winding number N is the number of times the skyrmion moments wrap the unit sphere.”

In addition to explicitly referring to the $\vec{G} \times \vec{v}$ term as the Magnus force, we have also replaced ∇U with F_{ext} . The result of these two changes is that discussions of different external forces (namely, the force due to spin-orbit torque) is now much easier to read, as discussed in response to the Referee's next comment.

Finally, the Referee states that “*Additionally, with respect to the helicity dependence of the skyrmion Hall angle for current induced experiments, the statement should be clarified as low alpha/high magnus force would mean a lower angle for Bloch skyrmions but still increase the mobility*”.

The skyrmion Hall angle is completely unaffected by the helicity dependence of the direction in which current-induced forces act. To be precise, irrespective of the direction in which \vec{F}_{ext} acts; the skyrmion Hall angle is always the angle between \vec{F}_{ext} and \vec{v} , as shown in figure 1(b). We hope that our reformulation of the discussion (detailed in the following reply), including an explicit example, helps to clarify this point.

On comment 5: “*The direction of motion of skyrmions can be further controlled by exploiting the direction in which external forces are applied. The spin-transfer torque effect applies a force to a skyrmion that is rotated by the helicity angle of the skyrmion. ...*”

I was more interested in the direct comparison of eq. 3+4 to the SOT/STT cases to make the reader aware of the different drive symmetries and thus angles they may observe in current driven experiments for Néel and Bloch cases under STT/SOT. As stated in the paragraph above, I think this aspect can be clarified a little more to avoid confusion.

Reply to 'On comment 5': We have modified our discussion to explicitly reformulate the Thiele equation for the case of SOT. The Reviewer rightly points out that there are different drive symmetries for Néel/Bloch type skyrmions driven by STT/SOT. A detailed discussion of these different drive symmetries is, however, the topic of an entire article (reference [7]). In order to keep the discussion concise, we mention only what one might consider to be the most interesting/confusing case: the case of a Bloch-type skyrmion driven by SOT. In this case, a Bloch-type skyrmion with a skyrmion Hall angle of 90° is driven parallel to the current density.

We agree that the business of driving Néel and Bloch type skyrmions using different techniques can be confusing, leading to lots of different drive directions. After discussing explicitly the Bloch-type skyrmion driven by SOT, we encourage readers to read reference [7] for a more in-depth discussion.

Additionally, the second sentence in the adjusted text reinforces once more that the skyrmion Hall angle is defined as the angle between an applied force (F_{ext} in Eq. (3)) and a skyrmion's resultant velocity – irrespective of the relationship between F_{ext} and parameters such as current density and helicity.

The amended paragraph now reads:

"The direction of motion of skyrmions can be further controlled by exploiting the direction in which external forces are applied. While the skyrmion Hall angle is defined as the angle between an applied force and a skyrmion's resultant velocity, the direction in which forces are applied can be complicated by the internal structure of skyrmions. In Eq. (3), the external force F_{ext} is set to be equal to the gradient of the Zeeman energy of a skyrmion. When F_{ext} is instead equated to the force due to spin-orbit torque from an adjacent heavy metal layer, the Thiele equation becomes

$$F_M - \alpha \vec{v} = kR(\phi)\vec{j}_{HM}$$

where k is a constant of proportionality, $R(\phi)$ is the rotation matrix in the plane spanned by F_m and \vec{v} , ϕ is the helicity angle of a skyrmion, and \vec{j}_{HM} is the current density in the heavy metal layer [7]. As Bloch-type skyrmions have a helicity of $\pm\pi/2$, the force due to spin-orbit torque is normal to \vec{j}_{HM} ; in this case, the direction of the skyrmion's induced velocity is parallel to \vec{j}_{HM} only when the skyrmion Hall angle is 90° . A detailed discussion of the relationship between current, force and drive direction for Néel- and Bloch-type skyrmions can be found in Ref. [7]."

Reviewer #3:

We thank the Reviewer once again for supporting the publication of our manuscript and for the helpful comments and suggestions, and we are glad that we have been able to address all of the referee's comments.

Below, we respond to the remaining comments.

Comment 1: *First, even if I agree that sputtering deposition is not the smoothest deposition technique but I would not have labelled it as "violent". The impact of this growth technique indeed depends on the materials that is deposited, as for example sputtered CoFeB films show extremely pinning site densities compared to Co ones. Second, because the authors have now introduced a specific argumentation about the drive dependence of the skyrmion Hall effect as well as a citation to several theoretical and experimental references, in order to be more exhaustive, they could probably also add the one published by W. Legrand et al, Nano Lett, 17, 2703 (2017).*

Reply to Comment 1: We have added the suggested reference, now Ref. [18]. We have also softened our wording of the discussion surrounding the sputtering deposition technique. We agree that the previous wording was too extreme. We have now rephrased that sentence to emphasize instead that sputtering is a highly energetic deposition technique, which, we hope you will agree, has a much more neutral tone. The adjusted sentence now reads:

"The highly energetic nature of the sputtering technique is expected to be the source of the pinning potentials present in these materials systems [17]."

Reviewers' Comments:

Reviewer #2:

Remarks to the Author:

The authors have adequately addressed all my previous points - especially the ones concerning the symmetry and definition of the skyrmion Hall angle (as angle between trajectory and driving force instead of the current direction as sometimes found in literature) for Bloch and Neel skyrmions. I have no further remarks and recommend publication.

Reviewer #3:

Remarks to the Author:

Report on paper: # NCOMMS-20-35818B by R. Brearton et al.

The authors have indeed improved their manuscript. However, they have only partially addressed the referees' comments, at least mine. Even if my opinion is that this article probably deserves to be published in Nature Communications, I still believe that it is important to answer to these questions and comments.

1) P5: as the skyrmion lattices in real systems are not ideal, the authors should comment what would be the consequences and if, for example, other defects could exist with other Burgers vectors and how this could modify the interpretation of the diffraction patterns?

4) P6, L125: The Gilbert damping coefficient is a key parameter for the estimation of the skyrmion Hall angle. Do the authors know how large it is for their FeGe film and what is the expected skyrmion Hall angle?

5) P7, L138, Fig4a,b : The authors must explain what are the side peaks present in Fig 4a and 4b. Also, these figures correspond to the diffraction pattern obtained under a perpendicular static field for stabilization of the skyrmion lattice. I guess that other diffraction patterns for lower fields, and hence in the helical phase, have been also recorded. It would be interesting to show them at least as supplementary materials.

6) P7 : L144, Fig. 4c: in Fig 4c, the diffraction diagram recorded in response to a shear force is presented. Even if it shows differences with the ones in Fig 4b, for example much less side peaks, I would not say that there is a striking difference between them as the angles of the six-fold peaks have only slightly changed by a few degrees. It is also not clear why the authors chose to wait for 15 min before recording the diffraction patterns. Is there any physical mechanism related to skyrmion physics associated to such a time scale?

7) P7, L149, Fig 4d : Why the beam stop direction has been changed for the experiments presented in Fig. b,c and the one in Fig 4d? Moreover, the authors use this diffraction pattern to estimate an angle between the direction of the applied force and the supposed direction of motion even though the experimental pattern (fig. 4d) is quite different from the calculated ones (Fig. 3b). In particular, in Fig. 4d, it is not so obvious to differentiate the elongated peaks from the ones that are expected to be punctual. The estimated skyrmion Hall angle is then estimated to be very large + 55° with a relatively small error bar. How this error bar has been determined?

8) P8: The authors must present at least in the supplementary materials the diffraction patterns obtained at different applied perpendicular fields. It would have been also interesting to display the diffraction patterns obtained for different field gradients if the experiments have been performed.

Reviewer #3:

We thank the Reviewer once again for supporting the publication of our manuscript.

Below, we respond to the remaining comments and answer the remaining questions.

Comment 1: *P5: as the skyrmion lattices in real systems are not ideal, the authors should comment what would be the consequences and if, for example, other defects could exist with other Burgers vectors and how this could modify the interpretation of the diffraction patterns?*

Reply to Comment 1: We believe that we have already addressed the comment in the previous round of reviews (previous Comment 2), and therefore direct the Reviewer to our answer given there. For convenience, we paste our answer below:

“With regards to your second point, it is true that edge dislocations can in general have Burgers vectors corresponding to any real lattice vector; 5-7 defects represent the special case that the Burgers vectors correspond to a primitive lattice vector. However, the magnitude of a Burgers vector denotes the magnitude of the corresponding lattice distortion. As can be seen by the Voronoi cells surrounding crystal points near the 5-7 defect in figure 2(a), the lattice is already significantly distorted by a Burgers vector of minimal length. Larger Burgers vectors correspond to extreme lattice distortions, which one would not expect to observe in experiment.

This has been confirmed by experiment, in e.g. Pöllath et al. [PRL 118, 207205 (2017)], where great numbers of 5-7 defects were measured using LTEM at domain boundaries. No higher order edge dislocations were reported. In order to strengthen our argument that one need only to consider 5-7 defects we have added a reference to Pöllath et al. (2017) in the main text.”

Comment 2: *P6, L125: The Gilbert damping coefficient is a key parameter for the estimation of the skyrmion Hall angle. Do the authors know how large it is for their FeGe film and what is the expected skyrmion Hall angle?*

Reply to Comment 2: As we discussed in the previous reply, the Gilbert damping coefficient is not the only important parameter for the estimation of the skyrmion Hall angle in metallic systems. As our previous discussion was perhaps insufficiently brief, we will now extend our argument.

The general form of Thiele's equation is given by

$$\mathbf{G} \times \mathbf{v} - \alpha \mathbf{v} = -\nabla U \quad (1)$$

where the parameter α can be written as

$$\alpha = \alpha_G \mathcal{D} + \alpha' \eta'$$

where the first term is linear in the coupling between the Gilbert damping and the dissipation tensor. The magnitude of the components of the dissipation tensor scale with skyrmion size (see part 2 of the supplementary of ref. [4] for analytic

approximations of its components). The second term arises due to coupling between a skyrmion and the conduction electrons.

Because a skyrmion is a topological object, when it moves, it appears to source an emergent electric field whose i^{th} component takes the value

$$E_{\text{Sk},i} = \frac{\hbar}{2q_e} \mathbf{m} \cdot \left(\partial_i \mathbf{m} \times \frac{d\mathbf{m}}{dt} \right)$$

where q_e is the charge of the electron, and the reduced Planck constant takes its usual symbol. This electric field gives rise to a current flowing from the skyrmion with magnitude

$$\mathbf{j} = \sigma \mathbf{E}_{\text{Sk}} \quad (2)$$

This current density exerts a spin-transfer torque on the magnetization of the crystal in which the skyrmion resides. The Landau-Lifshitz-Gilbert (LLG) equation including adiabatic and non-adiabatic spin-transfer torque terms can be written as

$$\frac{d\mathbf{m}}{dt} = -\gamma (\mathbf{m} \times \mathbf{H}_{\text{eff}} - \eta \mathbf{m} \times \frac{d\mathbf{m}}{dt}) + \frac{\hbar\gamma}{2q_e} \left((\mathbf{j}_{\text{Mag}} \cdot \nabla) \mathbf{m} - \beta \mathbf{m} \times (\mathbf{j}_{\text{Mag}} \cdot \nabla) \mathbf{m} \right)$$

where \mathbf{j}_{Mag} is the total current density flowing through the skyrmion-hosting material. Substituting Eq. (2) into the above expression for the LLG equation, premultiplying by $\mathbf{m} \times$, multiplying by $\partial_i \mathbf{m}$, and integrating gives equation (1), with

$$\eta' = \frac{1}{4\pi} \iint_S (\mathbf{m} \cdot (\partial_x \mathbf{m} \times \partial_y \mathbf{m}))^2 dx dy$$

and

$$\alpha' = (\hbar/2q_e)^2 \gamma \sigma$$

The above derivation was first carried out by Zang et al., PRL 107, 136804 (2011). It was also covered, in part, by Lin et al., (Ref. [9]). Here, formulae are presented with units consistent with the Thiele equation as derived in Ref. [9].

To calculate the conductivity of a lamella of FeGe would be a difficult and time-consuming process. A resultant estimate of the skyrmion Hall angle would inherit errors from the measurement of the Gilbert damping parameter, the conductivity of the sample, and the radial profile of the skyrmion, which is itself disputed (as the form of the magnetization determines the magnitude of both eta and the dissipative tensor). As the process of preparing lamellae by FIB is far from being a perfect, clean method, interactions with pinning potentials will likely also play a key role in the magnitude of the skyrmion Hall angle in our FeGe sample.

Consequently, as this paper aims to present a novel technique, we would rather avoid such off-topic technical details that would invariably provide little more than a poor estimate of the complicated parameter that is the skyrmion Hall angle.

Comment 3: P7, L138, Fig4a,b : *The authors must explain what are the side peaks present in Fig 4a and 4b. Also, these figures correspond to the diffraction pattern obtained under a perpendicular static field for stabilization of the skyrmion lattice. I guess that other diffraction patterns for lower fields, and hence in the helical phase, have been also recorded. It would be interesting to show them at least as supplementary materials.*

Reply to Comment 3: We direct the Reviewer to our answer to your previous comment.

As explained in the methods, figure 4(a) and (b) are averages over many experiments. There are many peaks because, over these experiments, the skyrmion lattice froze in a variety of different orientations.

Additionally, the caption for the figure itself refers to this scattering pattern as being a typical pattern for a skyrmion lattice with domains. There are no “side peaks”, only domains that are temporally separated as opposed to spatially separated. We also mention this point in our previous comment.

Comment 4: P7: L144, Fig. 4c: *in Fig 4c, the diffraction diagram recorded in response to a shear force is presented. Even if it shows differences with the ones in Fig 4b, for example much less side peaks, I would not say that there is a striking difference between them as the angles of the six-fold peaks have only slightly changed by a few degrees. It is also not clear why the authors chose to wait for 15 min before recording the diffraction patterns. Is there any physical mechanism related to skyrmion physics associated to such a time scale?*

Reply to Comment 3: We direct the Reviewer to our detailed response to your previous comment.

“Is there any physical mechanism related to skyrmion physics associated to such a time scale?”

The answer is yes, and there are two physical mechanisms to consider. Firstly, the initial orientation that the skyrmion lattice freezes into is random (biased by details such as sample geometry, field misorientation, anisotropy from the underlying crystal lattice etc.). It is possible that the initial orientation of the skyrmion lattice, including any domains, is not the most energetically preferential configuration of skyrmions. We measured that, over the timescale of ~minutes, minor lattice reorientation takes place as the skyrmion lattice attempts to find a configuration with a lower global energy.

The second effect leading to a timescale on the order of minutes is field gradient driven motion. In a previous proof of concept experiment, a skyrmion lattice was subject to a radial field gradient, which (for all nonzero skyrmion Hall angles) causes the skyrmion lattice to rotate. This rotation took place over the course of minutes – videos of the rotation can be found in the supplementary materials of S. L. Zhang et al., Nat. Commun. 9, 2115 (2018).

In order to control for the first physical mechanism (latent dynamics, minor reorientation), after entering the skyrmion pocket we waited for 15 minutes before driving the skyrmion lattice. We found that the dynamics of the skyrmion lattice in the absence of our drive were negligible. Once we started to drive the skyrmion lattice with a magnetic field gradient we applied the magnetic field gradient for 15 minutes,

because we know from Zhang [Nat. Commun. 9, 2115 (2018)] that this is a safe upper bound for the timescale of lattice dynamics induced by a magnetic field gradient.

“I would not say that there is a striking difference between them as the angles of the six-fold peaks have only slightly changed by a few degrees.”

We would have to agree: still figures simply don't do this effect justice – this is why we have attached supplementary videos of the transition from figure 4(b) to 4(c). The first supplementary video shows the result of one experiment, sped up by a factor of roughly 200. In that video, the effect of lattice reorientation in the presence of a field gradient is striking, even if the lattice only reorients by $\sim 10^\circ$.

The experiment shown in the first supplementary video was carried out 14 more times. Each time, the skyrmion lattice started at a different random initial orientation. Each time, the skyrmion lattice rotated to arrive at the same final orientation. The second supplementary video is an average over all these experiments. Of course, in this second video, there is no observable net rotation – the lattice rotates clockwise about as many times as it rotates counter-clockwise to reach its final orientation.

Finally, we note that the second supplementary video provides justification for our choice of timescale. A small amount of jiggling is noticeable throughout the “current off” phase, corresponding to the latent dynamics of the skyrmion crystal as it tries to find its optimal configuration. Also, once the field gradient is applied (corresponding to “current on” in the videos), it takes several seconds for all of the peaks from each of the experiments to become mutually aligned, meaning that it took a few minutes to guarantee lattice reorientation in the experiment.

In response to this point, we have explicitly referred to the supplementary videos in the caption for figure 4, to encourage the reader to watch for themselves the transition from figure 4(a) to (b) to (c).

Comment 5: *P7, L149, Fig 4d : Why the beam stop direction has been changed for the experiments presented in Fig. b,c and the one in Fig 4d? Moreover, the authors use this diffraction pattern to estimate an angle between the direction of the applied force and the supposed direction of motion even though the experimental pattern (fig. 4d) is quite different from the calculated ones (Fig. 3b). In particular, in Fig. 4d, it is not so obvious to differentiate the elongated peaks from the ones that are expected to be punctual. The estimated skyrmion Hall angle is then estimated to be very large $+ 55^\circ$ with a relatively small error bar. How this error bar has been determined?*

Reply to Comment 3: We direct the Reviewer to our answer to the Reviewer's previous comment, which we repeat for convenience below:

We thank the Reviewer for pointing this out, the beam stop direction had not changed. However, during the experiments at lower temperature, an alignment setting had been changed further down the beamline which lowered the intensity of the incident beam. This did not cause a practical problem as magnetic scattering was still very easy to resolve, however, it meant that a streak caused by previous damage to this particular CCD camera was on the same intensity order as magnetic scattering (and was therefore very bright). We have attached an un-edited, slightly cropped, version of figure 4(d) below:

The squares near the centre are caused by the direct beam passing through empty membranes in our sample holder – we had four membranes mounted at the time. The ugly mark on the right edge is the straight beam passing over the permanent magnets. In this larger image, it is hopefully easy to see that the beam stop was actually still horizontal, and that we added a second vertical one to mask the streak coming from detector damage that runs vertically through the screen.

If looking carefully at the image, one can see the remnants of this detector damage in the higher temperature experiments too, but as magnetic intensity was higher the streak shows up as a dark blue.

We decided that it is most reasonable to simply quote the angular width of the largest broadened diffraction spots in figure 4(d) as our greatest measurement error, as mentioned in the main text.

Comment 6: *P8: The authors must present at least in the supplementary materials the diffraction patterns obtained at different applied perpendicular fields. It would have been also interesting to display the diffraction patterns obtained for different field gradients if the experiments have been performed.*

As stated in our response to Comment 5, a repeat of the measurements of the other magnetic phases was not carried out given how precious (and expensive) synchrotron beamtime is. Instead, we refer to the work done by Birch et al. [Nat. Commun. 11, 1726 (2020)] on a very similar lamella cut from the very same crystal by the same person.

It was our intention for this first experiment to extract the skyrmion Hall angle that is as close to the maximum for our sample as possible. We do have data at lower drives, but it is limited and nowhere near as methodically obtained. It was captured while testing the maximum field gradient that we could safely apply before needing to consider effects due to heating. A study of the field-gradient dependence of these data would make for a very interesting follow-up experiment.